EMBO
Molecular Medicine

# 5-ALA does not potentiate dihydroartemisinin against *Plasmodium falciparum* malaria parasites

Hannah M Behrens [ID] [✉], Isabelle G Henshall [ID] & Tobias Spielmann [ID] [✉]

Artemisinin-derived drugs (ART) are fast-acting, efficient and cost-effective against malaria, but resistance against ART is spreading, necessitating improved ways of treatment. ART can also kill cancer cells, and this activity against cancer cells can be boosted by 5-aminolaevulinic acid (5-ALA) (Taubenschmid-Stowers et al, 2023). Hence, it might seem promising to use 5-ALA to potentiate ART efficiency and counteract the reduced ART-susceptibility of the parasite.

ART is activated by heme, which in cancer cells derives from the heme biosynthesis pathway. While heme biosynthesis can occur in infected red blood cells, *Plasmodium falciparum* malaria parasites release large quantities of heme from hemoglobin that they endocytose from the host red blood cell (Sigala and Goldberg, 2014). As a result, free heme is more abundant in *P. falciparum*-infected red blood cells, allowing for more efficient ART activation. This is likely why *P. falciparum* is several orders of magnitudes more susceptible to ART than cancer cells, with an IC50 of 0.7 nM for dihydroartemisinin (DHA, the active ART-metabolite) in *P. falciparum* (Witkowski et al, 2013) compared with 7.9 μM in glioblastoma cells (Taubenschmid-Stowers et al, 2023). *P. falciparum* parasites with Kelch13 mutations achieve decreased ART-sensitivity through decreased hemoglobin uptake in their ring stage, and thus decreased activation of ART, leading to partial ART-resistance (Behrens et al, 2021; Birnbaum et al, 2020; Zheng et al, 2024). 5-ALA is an intermediate in heme synthesis that can stimulate heme synthesis in the cells it is

taken up into. It is therefore possible that 5-ALA could potentiate ART by increasing ART activation to restore sensitivity in partially resistant parasites, as was recently suggested (Siddiqui et al, 2026).

ART-resistance only affects the ring stage of the intraerythrocytic parasite life cycle and therefore needs to be measured by a dedicated assay, the ring stage survival assay (RSA) (Witkowski et al, 2013). To investigate whether 5-ALA potentiates the effect of DHA, *P. falciparum* parasites harboring the resistance-conferring mutation K13 C580Y (Birnbaum et al, 2017), as well as their parental ART-sensitive 3D7 parasites, were exposed to DHA and 5-ALA during an RSA. In a first instance, parasites were co-exposed to 5-ALA and DHA for 6 hours (Fig. 1A) to observe whether RSA survival drops below the clinically relevant threshold of 1%. Two concentrations of 5-ALA were tested, which were selected based on their effect on cancer cells: an intermediate concentration of 200 μM and the maximum concentration of 1000 μM (Taubenschmid-Stowers et al, 2023). At both concentrations, more than 20% of parasites harboring the resistance-conferring K13 C580Y mutation survived, and this was in a similar range to the control without 5-ALA (Fig. 1A). This indicated that 5-ALA co-treatment had no or only a minimal effect and was not sufficient to overcome the reduced susceptibility of parasites harboring K13 C580Y.

To explore this more thoroughly and to maximize 5-ALA impact, we next preincubated the parasites with 1000 μM 5-ALA in the 3 h of the RSA during which the parasites invade new red blood cells prior to the DHA-treatment and maintained

5-ALA exposure during the 6 h DHA-treatment, resulting in a total of 9 h of 5-ALA incubation. While survival in the 5-ALA incubated parasites was slightly reduced, this effect on the survival of K13 C580Y parasites was not statistically significant and did not reduce the proportion of surviving parasites below 1% (Fig. 1B). Overall, 5-ALA was not found to be suitable to counteract the reduced susceptibility to DHA caused by the K13 mutation C580Y.

We also tested whether 5-ALA potentiated the effect of DHA on susceptible 3D7 parasites. To be able to observe a possible effect, the DHA concentration was reduced to 3 nM, at which 5–20% of parasites survive the DHA exposure in the RSA. The susceptible parasites were incubated with 5-ALA for 9 h as before. Again, survival between 5-ALA-treated and untreated parasites was comparable (Fig. 1C).

In all experiments, we observed a small decrease in parasite survival in the presence of 5-ALA, and this may correspond to the mild effect observed by Wang and colleagues in 5-ALA-treated 3D7 parasites (Wang et al, 2015). Nonetheless, in conclusion, 5-ALA was not able to overcome the reduced ART-susceptibility observed in Kelch13-mutant parasites, nor did it improve the effect of DHA against susceptible *P. falciparum* parasites in a way expected to be clinically useful.

The uptake of 5-ALA into erythrocytes during the trophozoite stage is dependent on a parasite-derived channel on the erythrocyte surface, the Plasmodial Surface Anion Channel PSAC, that is not functional during the ring stage (Sigala et al, 2015). To investigate whether the failure of 5-ALA to

Malaria Cell Biology, Molecular Biology and Immunology, Bernhard Nocht Institute for Tropical Medicine, Bernhard-Nocht-Str. 74, 20359 Hamburg, Germany.
✉E-mail: hannah.behrens@bnitm.de; spielmann@bnitm.de
https://doi.org/10.1038/s44321-026-00388-7 | Published online: 27 February 2026

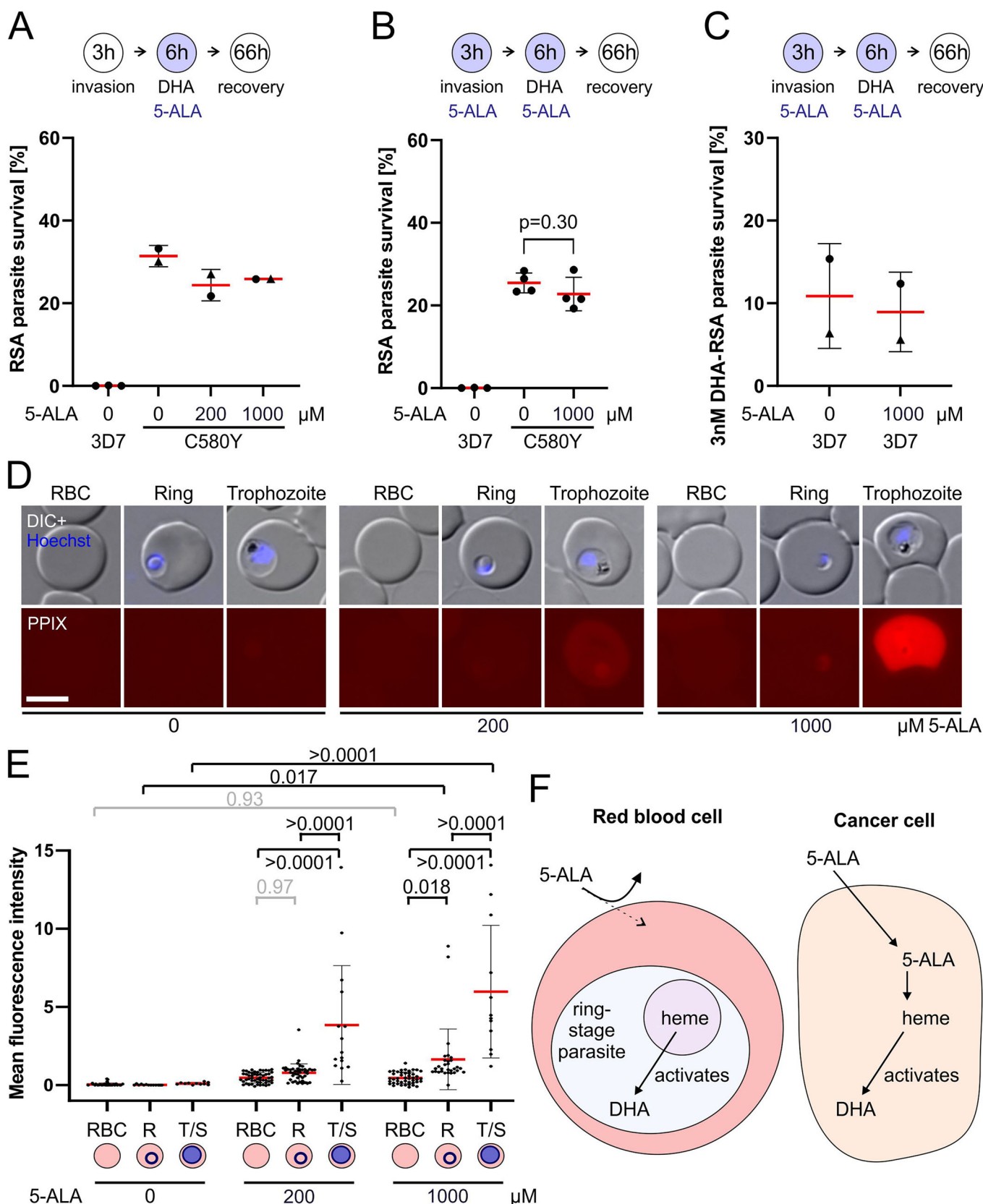

**Figure 1. 5'ALA does not potentiate DHA in its cytotoxic activity against *P. falciparum* parasites.**

(A) RSA survival of resistant parasites harboring Kelch13 mutation C580Y with 0, 200, or 1000 µM 5-ALA added during the assay compared with susceptible parental 3D7 parasites without 5-ALA exposure. The standard concentration of 700 nM DHA was used. 5-ALA was added during the DHA incubation, resulting in a 6 h co-incubation (scheme). For 3D7 $n = 3$, for K13 C580Y $n = 2$ where data points represented by the same symbols derive from experiments on the same days. (B) RSA survival as in (A) but with 0 or 1000 µM 5-ALA added during the 3 h invasion prior to DHA addition and during the 6 h DHA incubation, resulting in a total of 9 h 5-ALA exposure. For 3D7 $n = 3$, for K13 C580Y $n = 4$. $P$ value from an unpaired Students's $t$-test. (C) RSA survival of ART-sensitive 3D7 parasites with 0, 200 or 1000 µM 5-ALA added as in (B), resulting in a total of 9 h 5-ALA exposure. A lower-than-usual concentration of 3 nM DHA was used to allow the survival of a proportion of these DHA-sensitive parasites. $n = 2$. Data points represented by the same symbols derive from experiments on the same days. (D) Representative microscopic images of uninfected and infected red blood cells incubated with 0, 200, or 1000 µM 5-ALA, showing DIC and Hoechst staining and red PPIX fluorescence indicative of 5-ALA uptake. Size bar 5 µM. DIC differential interference contrast. (E) Quantification of red fluorescence indicative of 5-ALA uptake in uninfected and infected red blood cells incubated with 0, 200, or 1000 µM 5-ALA. RBC, red blood cell; R, rings; T/S, trophozoites and schizonts. Relevant $p$ values from one-way ANOVA are shown, and colored in black ($p < 0.05$) or gray ($p > 0.05$). (F) Working model of 5-ALA uptake and ART activation in *P. falciparum*-infected red blood cells and cancer cells. (A–D) Red bars show the mean, and error bars show the standard deviation. $n$ is defined as the number of independent experiments. Groups of two were analyzed with Student's $T$-test, and more were analyzed by one-way ANOVA. Source data are available online for this figure.

overcome ART-resistance was due to inefficient uptake of 5-ALA into the parasite, we monitored 5-ALA uptake by fluorescence microscopy. As part of heme biosynthesis, 5-ALA can be converted to Protoporphyrin IX (PPIX), which is fluorescent. While trophozoite and schizont parasites efficiently took up 5-ALA, there was only limited PPIX fluorescence in rings (Figure D+E). Thus, either the limited uptake of 5-ALA or insufficient conversion of 5-ALA to PPIX and heme is the reason why 5-ALA cannot restore sensitivity to parasites in the resistance-relevant ring stage of the parasite (Fig. 1D–F). Some brain tumors, on the other hand, accumulate 5-ALA (Stummer et al, 1998), and in these cells, heme synthesis is the main source of ART activation, which is why 5-ALA potentiates DHA in these tissues (Taubenschmid-Stowers et al, 2023). While it might be possible to overcome the poor uptake into infected red blood cells through the development of a more membrane-permeable derivative of 5-ALA in the future, it is unclear if a sufficient amount of such a 5-ALA derivative can be converted to heme to increase ART activation in rings. Hence, swift repurposing of human use-approved 5-ALA does not appear to be a viable strategy.

## Methods

### Reagents and tools table

| Reagent/ resource | Reference or source | Identifier or catalog number |
|---|---|---|
| **Experimental models** | | |
| *P. falciparum* 3D7 | Walliker et al, 1987 | |
| *P. falciparum* 3D7 K13 C580Y | Birnbaum et al, 2017 | |

| Reagent/ resource | Reference or source | Identifier or catalog number |
|---|---|---|
| **Recombinant DNA** | | |
| NA | | |
| **Antibodies** | | |
| NA | | |
| **Oligonucleotides and other sequence-based reagents** | | |
| NA | | |
| **Chemicals, enzymes and other reagents** | | |
| Albumax | Life Technologies | Gibco 11021-045 |
| DHA | AdipoGen Life Sciences | AG-CN2-0468-M050 |
| 5-ALA | Sigma | A7793-10MG |
| **Software** | | |
| Graphpad Prism | www.graphpad.com/ | |
| Corel V7 | www.corel.com/en/ | |
| **Other** | | |

*P. falciparum* 3D7 (Walliker et al, 1987) and K13 C580Y parasites (Birnbaum et al, 2017) were cultivated as described previously (Trager and Jensen, 1976) in 5% hematocrit human 0+ erythrocytes at 37 °C in RPMI complete medium with 0.5% Albumax (Life Technologies). All parasites were mycoplasma-negative. RSA were performed to assess DHA susceptibility by the standardized procedure described previously (Witkowski et al, 2013). 0-h- to 3-h-old ring-stage parasites were exposed to 700 nM DHA for 6 h and subsequently cultivated for 66 h. Parasite survival rate was determined by Giemsa smear

comparing parasitemia of viable parasites after DHA exposure to that of the untreated control. The procedure was modified for selected samples by adding 5-ALA (Sigma, A7793-10MG) three hours prior to DHA exposure and during the six hours of DHA exposure. 3 nM instead of 700 nM DHA was used for the 6 h DHA exposure in some samples, as indicated. 5-ALA uptake was investigated by 16 h incubation of asynchronous parasites with 5-ALA, followed by fluorescence microscopy with a Zeiss Axioimager M1 with a 64 HE filter. All images were taken at the same settings and modified for intensity in identical ways. For quantification, intensity was not adjusted. Samples were blinded for analysis.

## Data availability

This study includes no data deposited in external repositories.

## Peer review information

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

## Acknowledgements

The authors are grateful for funding from the European Research Council (ERC) under the European Union's Horizon 2020 research and innovation program (grant agreement number 101021493). We thank Tim Wells (Medicines for Malaria Venture) for proposing to test this after a prompt from Moritz Horn (JLP Health GmbH), who tested 5-ALA in cancer cells.

## Author contributions

**Hannah M Behrens**: Conceptualization; Formal analysis; Investigation; Visualization; Writing—original draft; Writing—review and editing. **Isabelle G Henshall**: Investigation; Writing—review and editing. **Tobias Spielmann**: Conceptualization; Funding acquisition; Writing—review and editing.

## Disclosure and competing interests statement

The authors declare no competing interests.

