## [Peer Review File · EMBO Molecular Medicine]

5-ALA does not potentiate dihydroartemisinin against *Plasmodium falciparum* malaria parasites

Hannah Behrens, Isabelle Henshall, and Tobias Spielmann

Corresponding authors: Hannah Behrens (hannah.behrens@bnitm.de) , Tobias Spielmann (spielmann@bnitm.de)

Review Timeline:

Submission Date:	20th Aug 25
Editorial Decision:	10th Sep 25
Revision Received:	18th Dec 25
Accepted:	22nd Jan 26

Editor: Zeljko Durdevic

Transaction Report:

10th Sep 2025

Dear Dr. Behrens,

Thank you for the submission of your manuscript to EMBO Molecular Medicine. We have now received feedback from a reviewer who agreed to evaluate your manuscript. As you will see from the report pasted below, the referee recognizes potential interest of the study but also raises a number of concerns that should be addressed in a major revision. Please perform the additional experiment and discuss your findings considering the preprint and the published manuscript as suggested by the referee.

Furthermore, your manuscript would be suitable for further consideration only as a Correspondence article. A Correspondence has no abstract and should be limited to 1000 words in principle. The number of references should not exceed 10. Please check recently published Correspondence articles copied below for the formatting reference and our Author Guidelines for more information about manuscript preparation.

<https://www.embopress.org/doi/full/10.1038/s44321-024-00044-y>
<https://www.embopress.org/doi/full/10.1038/s44321-024-00063-9>
<https://www.embopress.org/page/journal/17574684/authorguide#correspondenceguide>

EMBO Molecular Medicine encourages a single round of revision only and therefore, acceptance or rejection of the manuscript will depend on the completeness of your responses included in the next, final version of the manuscript. For this reason, and to save you from any frustrations in the end, I would strongly advise against returning an incomplete revision.

Please also amend the following:

Submit a complete Author Checklist. <https://www.embopress.org/pb-assets/embo-site/EMBO%20Press%20Author%20Checklist-1642513524327.xlsx>

Remove the figure from the main manuscript file and upload it as a separate high-resolution file. Figure legend should be placed after references.

We would welcome the submission of a revised version within three months for further consideration. Please let us know if you require longer to complete the revision.

I look forward to receiving your revised manuscript

Yours sincerely,

Zeljko Durdevic

Zeljko Durdevic
Senior Editor
EMBO Molecular Medicine

*** Instructions to submit your revised manuscript ***

- 1) a .docx formatted version of the manuscript text (including Figure legends and tables)
- 2) Separate figure files*
- 3) supplemental information as Expanded View and/or Appendix. Please carefully check the authors guidelines for formatting Expanded view and Appendix figures and tables at <https://www.embopress.org/page/journal/17574684/authorguide#expandedview>
- 4) a letter INCLUDING the reviewer's reports and your detailed responses to their comments (as Word file).
- 5) EMBO Molecular Medicine now requires a complete author checklist (<https://www.embopress.org/page/journal/17574684/authorguide>) to be submitted with all revised manuscripts. Please use the checklist as guideline for the sort of information we need WITHIN the manuscript. The checklist should only be filled with page numbers where the information can be found. This is particularly important for animal reporting, antibody dilutions (missing) and exact values and n that should be indicated instead of a range.
- 6) A Disclosure and competing interest statement should be provided in the main text
- 7) Please note that we now mandate that all corresponding authors list an ORCID digital identifier. This takes <90 seconds to complete. We encourage all authors to supply an ORCID identifier, which will be linked to their name for unambiguous name identification.

Currently, our records indicate that the ORCID for your account is 0000-0002-7406-7086.

Link Not Available

Photos 400-800 DPI

*Additional important information regarding figures and illustrations can be found at

<https://bit.ly/EMBOPressFigurePreparationGuideline>. See also figure legend preparation guidelines:

<https://www.embopress.org/page/journal/17574684/authorguide#figureformat>

***** Reviewer's comments *****

Referee #1 (Remarks for Author):

Artemisinin (ART) is a frontline antimalarial drug whose endoperoxide functionality requires reductive activation inside cells via interactions with ferrous heme. ART has shown some efficacy against certain cancers, and its activity can be increased by stimulating heme synthesis via exogenous provision of 5-ALA, which is a chemical intermediate in heme synthesis and stimulates biosynthesis of protoporphyrin IX (PPIX) and (depending on iron availability) heme.

In the present manuscript, the authors test whether ALA can enhance ART activity against blood-stage *P. falciparum*, using pulsed 6-hr drug exposure and ring-stage survival assay (RSA), which is the parasite stage associated with ART tolerance in Kelch-13 protein mutants. The authors find no evidence that ALA enhances the sensitivity of ring-stage parasites to ART, including both ART-sensitive and ART-tolerant parasites. This observation is consistent with the consensus view that host

hemoglobin-derived heme, rather than biosynthetic heme, is the dominant activator of ART in blood-stage Plasmodium.

Prior work (Sigala et al. study cited by authors) showed that ALA uptake requires parasite-induced permeability pathways in the host RBC that are turned on in late rings. Given known RBC impermeability to ALA before induction of NPPs, the rationale for expecting that ALA would enhance ring-stage parasite sensitivity to ART is unclear. Thus, the result of this study is unsurprising based on what was already known about the permeability of parasite-infected RBCs to ALA. The authors' results are clear, but they could consider a simple microscopy experiment to test and show that 6-hr ALA treatment of ring-stage parasites does not result in detectable PPIX fluorescence, unlike in trophozoites and schizonts (as performed in Sigala et al.).

Despite low expectation that ALA would enhance ART activity against ring-stage parasites, there is a recent preprint that concluded that ALA does increase ART activity against blood-stage Plasmodium (<http://dx.doi.org/10.21203/rs.3.rs-4535885/v1>). This preprint, which appears to be poorly controlled and of low quality based on a quick scan, was not cited by the authors as motivation for their study.

Nevertheless, there may be value in publishing a properly controlled study, like the present manuscript, to de-muddle the literature on this point. The authors may also wish to note that their results contrast with a prior highly-cited manuscript that suggested the heme biosynthesis was a dominant activator of ART in ring-stage parasites (<https://www.nature.com/articles/ncomms10111>).

***** Reviewer's comments *****

We very much thank the reviewer for the constructive suggestions and criticism which helped us to improve the manuscript.

Referee #1 (Remarks for Author):

Artemisinin (ART) is a frontline antimalarial drug whose endoperoxide functionality requires reductive activation inside cells via interactions with ferrous heme. ART has shown some efficacy against certain cancers, and its activity can be increased by stimulating heme synthesis via exogenous provision of 5-ALA, which is a chemical intermediate in heme synthesis and stimulates biosynthesis of protoporphyrin IX (PPIX) and (depending on iron availability) heme.

In the present manuscript, the authors test whether ALA can enhance ART activity against blood-stage *P. falciparum*, using pulsed 6-hr drug exposure and ring-stage survival assay (RSA), which is the parasite stage associated with ART tolerance in Kelch-13 protein mutants. The authors find no evidence that ALA enhances the sensitivity of ring-stage parasites to ART, including both ART-sensitive and ART-tolerant parasites. This observation is consistent with the consensus view that host hemoglobin-derived heme, rather than biosynthetic heme, is the dominant activator of ART in blood-stage *Plasmodium*.

Prior work (Sigala et al. study cited by authors) showed that ALA uptake requires parasite-induced permeability pathways in the host RBC that are turned on in late rings. Given known RBC impermeability to ALA before induction of NPPs, the rationale for expecting that ALA would enhance ring-stage parasite sensitivity to ART is unclear.

Reply: We acknowledge that PSAC is an important factor for 5ALA uptake. It has been shown that uninfected red blood cells and red blood cells infected with trophozoite stage parasites in which NPP activity has been blocked are impermeable to ALA (Sigala et al.). However, to our knowledge, RBC permeability to ALA in ring stages has not been studied.

Thus, the result of this study is unsurprising based on what was already known about the permeability of parasite-infected RBCs to ALA. The authors results are clear, but they could consider a simple microscopy experiment to test and show that 6-hr ALA treatment of ring-stage parasites does not result in detectable PPIX fluorescence, unlike in trophozoites and schizonts (as performed in Sigala et al.).

Reply: We have now performed the suggested microscopy experiment to provide the first insights into 5-ALA uptake in ring stage parasites. Perhaps surprisingly, we found that while the PPIX fluorescence, indicative of 5-ALA uptake, is much lower in ring-infected RBCs than in trophozoite- and schizont-infected RBCs, some 5-ALA does enter ring-infected RBCs, as PPIX fluorescence is higher than in uninfected RBCs. While the uptake route is unclear, this fits with the small non-significant decrease in DHA-survival that we observe (Figure 1A-C) and with the small decrease in DHA-survival of ring stages that Wang et al observed (over a shorter time span and lower DHA concentration). The fact that this is effect is small also fits with our observation that the effect on DHA-survival is not large enough to overcome ART-resistance (Figure 1 D+E).

Despite low expectation that ALA would enhance ART activity against ring-stage parasites, these is a recent preprint that concluded that ALA does increase ART activity against blood-stage Plasmodium (<http://dx.doi.org/10.21203/rs.3.rs-4535885/v1>). This preprint, which appears to be poorly controlled and of low quality based on a quick scan, was not cited by the authors as motivation for their study.

Reply: This study is now cited as additional motivation.

Nevertheless, there may be value in publishing a properly controlled study, like the present manuscript, to de-muddle the literature on this point. The authors may also wish to note that their results contrast with a prior highly-cited manuscript that suggested the heme biosynthesis was a dominant activator of ART in ring-stage parasites (<https://www.nature.com/articles/ncomms10111>).

Reply: In the mentioned study by Wang et al. Nature Communications 2015, Wang et al. perform experiments to investigate whether heme from heme biosynthesis or from endocytosis is the dominant activator of ART in ring-stages, concluding that heme biosynthesis is dominant. This is in part based on the observation that blocking heme biosynthesis using the inhibitor SA increased ART survival by ~5% while an inhibitor of endocytosis, ALLN, showed no effect. It is not clear whether ALLN was in fact inhibiting endocytosis in these experiments or whether it had for instance no effect due to a lack of uptake into ring-infected red blood cells. In the ten years since the study by Wang et al. was published, new evidence has revealed that endocytosis does in fact take place in ring stages and that reduction of endocytosis increases ART survival, presumably through reduced activation of ART (Klonis et al. 2011 PMID: 21709259, Xie et al. 2016 PMID: 26675237, Yang et al. 2019 PMID: 31775055, Birnbaum et al. 2020 PMID: 31896710, Behrens et al. 2024 PMID: 34309894).

In our study, we report limited PPIX fluorescence in ring-infected RBCs after 5-ALA exposure and a limited effect on ART susceptibility in ring stage parasites. It is currently unclear whether the limiting factor is 5-ALA uptake or conversion of 5-ALA into detectable PPIX and subsequently heme.

However, as mentioned above, Wang et al. observed a small effect of ALA on DHA-susceptibility, which agrees with our observations. By using the standardized RSA protocol, for which clinically relevant survival rates have been established (Witkowski et al., 2013 PMID: 23208708), we were able to conclude that this effect is not sufficient to overcome K13 mutation-mediated ART resistance. We have added a discussion of the results concerning this effect observed by Wang et al as far as possible to remain within the format of a correspondence.

22nd Jan 2026

Dear Dr. Behrens,

We are pleased to inform you that your manuscript is accepted for publication and is now being sent to our publisher to be included in the next available issue of EMBO Molecular Medicine.

You may qualify for financial assistance for your publication charges - either via a Springer Nature fully open access agreement or an EMBO initiative. Check your eligibility: <https://link.springer.com/journal/44321/how-to-publish-with-us>

Zeljko Durdevic
Senior Editor
EMBO Molecular Medicine

>>> Please note that it is EMBO Molecular Medicine policy for the transcript of the editorial process (containing referee reports and your response letter) to be published as an online supplement to each paper. If you do NOT want this, you will need to inform the Editorial Office via email immediately. More information is available here: <https://link.springer.com/partners/embo-press/editorial-policies#Peer%20review>